# MYB and ELF3 differentially modulate labor-inducing gene expression in myometrial cells

**Virlana M. Shchuka**[1]\*, **Nawrah Khader** [1], **Anna Dorogin**[2,3], **Oksana Shynlova**[2,3]\*, **Jennifer A. Mitchell** [1]\*

1 Department of Cell and Systems Biology, University of Toronto, Toronto, Ontario, Canada, 2 Lunenfeld Tanenbaum Research Institute, Sinai Health System, Toronto, Ontario, Canada, 3 Department of Obstetrics and Gynaecology, University of Toronto, Ontario, Canada

\* virlana.shchuka@mail.utoronto.ca (VMS); ja.mitchell@utoronto.ca (JAM); shynlova@lunenfeld.ca (OS)

## Abstract

Spontaneous uterine contractions are initiated when smooth muscle cells (SMCs) within the uterine muscle, or myometrium, transition from a functionally dormant to an actively contractile phenotype at the end of the pregnancy period. We know that this process is accompanied by gestational time point-specific differences in the SMC transcriptome, which can be modulated by the activator protein 1 (AP-1), nuclear factor kappa beta (NF-κβ), estrogen receptor (ER), and progesterone receptor (PR) transcription factors. Less is known, however, about the additional proteins that might assist these factors in conferring the transcriptional changes observed at labor onset. Here, we present functional evidence for the roles of two proteins previously understudied in the SMC context—MYB and ELF3—which can contribute to the regulation of labor-driving gene transcription. We show that the *MYB* and *ELF3* genes exhibit elevated transcript expression levels in mouse and human myometrial tissues during spontaneous term labor. The expression of both genes was also significantly increased in mouse myometrium during preterm labor induced by the progesterone antagonist mifepristone (RU486), but not during infection-simulating preterm labor induced by intrauterine infusion of lipopolysaccharide (LPS). Furthermore, both MYB and ELF3 proteins affect labor-driving gene promoter activity, although in surprisingly opposing ways: *Gja1* and *Fos* promoter activation increases in the presence of MYB and decreases in the presence of ELF3. Collectively, our study adds to the current understanding of the transcription factor network that defines the transcriptomes of SMCs during late gestation and implicates two new players in the control of labor timing.

## Introduction

As the gestational period progresses, smooth muscle cells (SMCs) in the myometrium, or muscular component of the uterus, undergo several consecutive phases of structural change. One such change occurs near the end of pregnancy, when a substantial portion of SMCs adopt a state of contractile activity and promote the onset of labor. The difference in contractile ability between the functionally dormant cells at early gestation and those at full term ($\geq$ 37 weeks in humans) is reflected through distinct, context-specific SMC transcriptomes. RNA-seq profiles

**Funding:** This work was supported by the Canadian Institutes of Health Research (FRN 173252, held by J.A.M. and O.S.; cihr-irsc.gc.ca). Studentship funding was provided by the Natural Science and Engineering Research Council of Canada (CGS D held by V.M.S.; nserc-crsng.gc.ca). The funders had no role in study design, data collection and analysis, decision to publish, or preparation of the manuscript.

**Competing interests:** The authors have declared that no competing interests exist.

generated from human myometrial cells indicate that these disparate gene expression patterns are determined by such conditions as: whether SMCs are isolated from a non-pregnant or pregnant individual [1]; labor status at term, or whether term-isolated SMCs were engaged in induced labor or active, spontaneous labor at the time of isolation [1, 2]; the time point of labor, or whether or not SMCs are obtained at term or in advance of term [3–5]; and, in some cases, the relative localization within the myometrium or sub-population of the isolated SMCs [3, 6]. In rodent models of labor, where SMCs can be collected from a wider range of gestational time points than their human counterparts, distinct stage-specific gene expression profiles are also observed. Myometrial cells isolated in the late gestational period, for example, exhibit substantially different transcriptomes from those isolated at term labor [7, 8]. Furthermore, the SMC transcriptome can differ depending on the particular mode of labor. Spontaneous term labor and experimentally induced preterm labor by either non-infectious or infection-simulating inflammation in mice show a degree of overlap in commonly expressed genes; however, these common mouse models of idiopathic preterm labor and infection-induced preterm labor respectively yield contractile SMCs with non-identical gene expression profiles [7].

Our previous work revealed that the majority of gene expression differences between late gestation SMCs and contracting SMCs in the mouse myometrium at term occurs due to changes in active transcription events. More specifically, many contraction-driving genes undergo increased primary transcript synthesis at labor onset [8]. For instance, the widely studied Connexin 43 (encoded by the *GJA1* gene), a vital contraction-associated protein [9, 10], is expressed at low levels in the late gestational quiescent stages, and sharply induced at term labor onset in part due to the spike in active transcription at its coding gene [8]. But the mechanism by which these transcriptional changes are conferred on genes like *GJA1* during the SMC quiescence-to-contractility transition is not well understood. The control of patterns of transcriptional activation directly relies on the formation of complexes comprised of key transcription factor proteins, which operate in the same or co-existing regulatory circuitries. Some of the more extensively studied transcription factors that have been linked to the labor-driving transcriptional changes observed in SMCs include members of the Activator protein 1 (AP-1), Nuclear factor kappa beta (NF-κβ), Estrogen receptor (ER), and Progesterone receptor (PR) families. For example, AP-1 heterodimers, but not homodimers, tend to activate the murine *Gja1* promoter [11], and this heterodimeric activation effect can be heightened or hampered depending on which major PR isoform is present in SMCs [12].

More recent studies have shown preliminary evidence of co-operation between the more well-established transcription factor quartet and other candidate factors less well-known in this context, such as members of the KLF, SMAD, FOXO, IRF, and HOXA families (reviewed in [13]). Despite these initial advances in identifying putative members of gestational stage-specific transcription factor complexes, the full repertoire of proteins that contribute to the laboring phenotype in SMCs is not comprehensively characterized. In this paper, we present evidence for two additional proteins that contribute to the downstream control of gene regulation in these cells: Myeloblastosis proto-oncogene (MYB) and E74 like ETS transcription factor 3 (ELF3). We hypothesized that these two proteins might act as potential regulators of the laboring SMC phenotype for several reasons: both their corresponding genes are more highly transcribed at term relative to earlier quiescent gestational stages in mouse myometrial tissues [7, 8, 14]. Associated predicted binding motifs for both proteins are also contained within promoters of contraction-associated genes like *Gja1*. More notably, MYB and ELF3 have been shown to behave as transcriptional activators in cooperation with AP-1 factors in other cellular contexts. For example, Tapias and others determined that, in HeLa cells, MYB controls the transcription of and co-localizes with AP-1 factors at the promoter of *Sp3* [15], a gene

prominently expressed in SMCs [reviewed in 13]. Likewise, a study by Otero and others demonstrated that ELF3 binds the *Mmp13* gene promoter in chondrocyte cells; while AP-1 heterodimers can activate the *Mmp13* promoter, the addition of ELF3 further compounds this activation effect [16]. ELF3 has also been shown to interact with NF-κβ in prostate cancer cells in response to pro-inflammatory stimuli that also affect late-gestation SMCs [17]. In light of these findings, we hypothesized that MYB and ELF3 might be significant members of the laboring SMC transcription regulatory network, and ones that function in conjunction with AP-1 factors.

Here, we demonstrate that, as in the case of many labor-driving candidate genes, *MYB* and *ELF3* follow a differential expression pattern trend in mice and humans as SMCs transition from late pregnancy to labor onset. We also observed these expression differences in a mouse model of idiopathic preterm labor, but not in a model of infection-induced preterm labor. Furthermore, we explore the functional potential of these factors in driving the regulation of labor-associated genes. Somewhat surprisingly, we observed opposing effects from each of these factors on labor-associated gene promoter stimulation. Our work here highlights the abilities of MYB and ELF3 to contribute in different ways to the labor-associated gene expression program in myometrial cells. In so doing, we expand the current repertoire of proteins associated with SMC state definition during the gestational period and offer two potential new targets for the development of therapies toward labor timing-related health challenges.

## Results

### Term labor onset is associated with increased expression of MYB and ELF3 factors in murine and human myometrial tissues

To pinpoint candidate factors that might contribute to the transcriptional up-regulation of labor-associated genes, we considered genes encoding transcription factors that are differentially expressed between gestational days 15 and 19 (active labor) from our previously generated mouse gestational RNA-seq dataset [8]. The expression levels of two labor-upregulated gene targets, *Myb* and *Elf3*, were examined by RT-qPCR from RNA extracted at select late gestational time points in both murine and human myometrial tissues (Fig 1A). Across the late gestational timeline, we observed in CD-1 and Bl6 murine models that both Myb and Elf3 transcript levels significantly increase by a respective factor of 3- and 5-fold toward term (Fig 1B; **S1 Fig in** S1 File). Human myometrial cells exhibit a similar profile *in vivo*, where transcript levels corresponding to both *MYB* and *ELF3* are substantially higher during active term labor relative to term non-laboring samples (Fig 1C). The sharp, stage-specific expression increase of these transcripts at this particular time point of myometrial cell state transition suggests that these genes may encode critical regulators of the laboring myometrial cell phenotype.

### Significant elevation in Myb and Elf3 expression levels is observed in an idiopathic mouse model of preterm labor but not in an infection-associated model

Having observed the elevated expression of the *Myb* and *Elf3* genes from late pregnancy to term labor, we next sought to establish whether these genes followed a similar activation trend in mouse preterm labor models. We investigated levels of Myb and Elf3 transcripts using two established experimental models known to prematurely trigger labor onset in mice: by *E. coli*-derived lipopolysaccharide (LPS) treatment to mimic infection-induced uterine inflammation; and by mifepristone (RU486) treatment (which blocks the progesterone signaling pathway) to

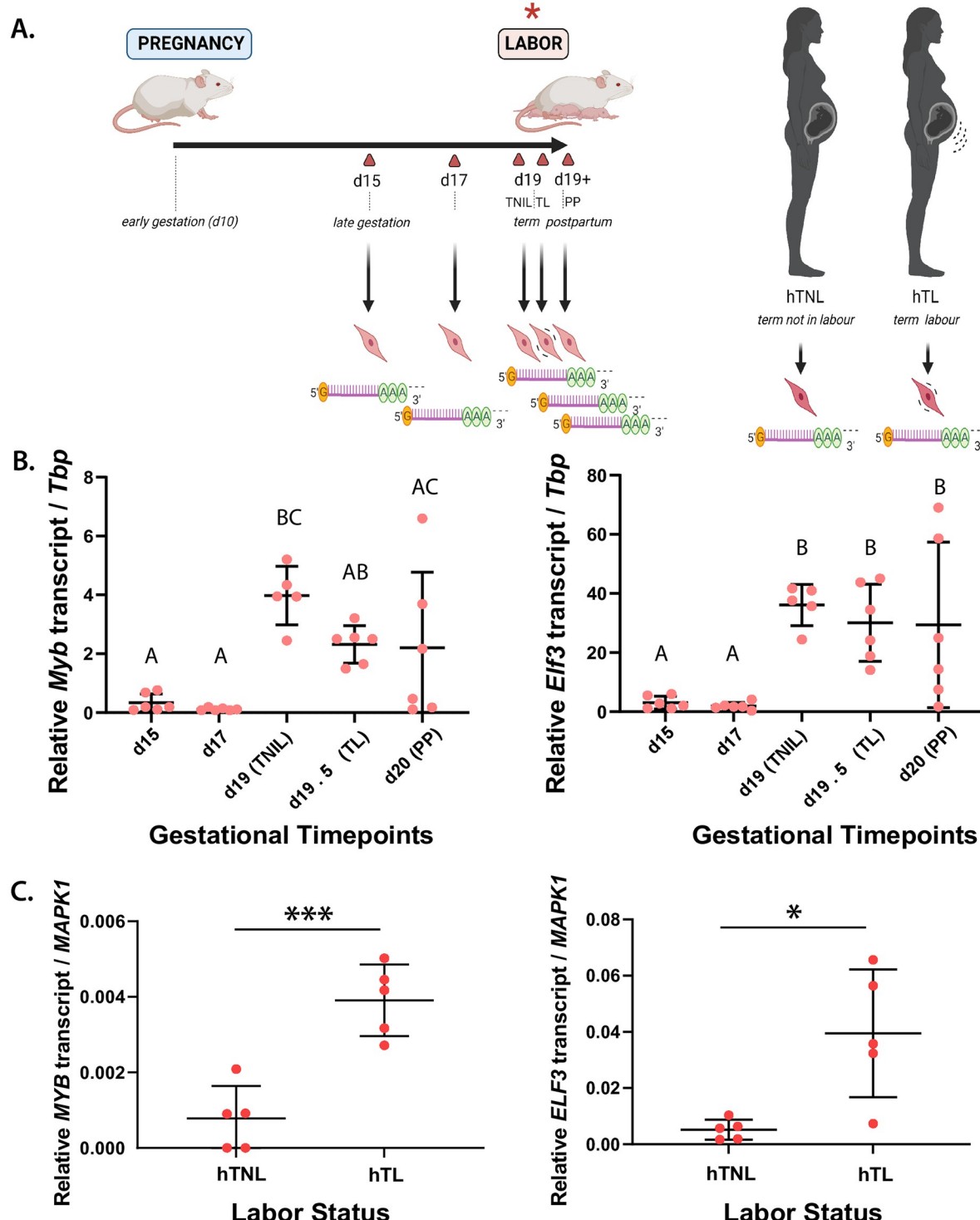

**Fig 1.** *MYB* and *ELF3* are differentially expressed in CD-1 mouse and human myometrial tissues *in vivo* depending on labor status.
**(A)** *(left)* Schematic of gestational time points for myometrial tissue collection from CD-1 mice. Gestational days (d) at which tissues were collected marked by triangles (red), with indication of term non-laboring (TNIL), term laboring (TL) and postpartum (PP) mouse myometrium alongside key gestational stages *(italicized)*. *(right)* Schematic of RNA collection from human myometrial tissues, from either the term non-laboring (hTNL) or term laboring (hTL) women. **(B)** Expression levels (±S.D.) of transcription factor-encoding genes at d15, d17, d19 (TNIL), d19.5 (TL), and d20 (PP) time points, as determined by RT-qPCR. Groups determined by one-way ANOVA to be significantly different (p < 0.05) from one another are labeled with different letters; to indicate p > 0.05, groups are labeled with the same letter. **(C)** Expression levels (±S.D.) of transcription factor-encoding genes in human TNL and TL myometrium, as determined by RT-qPCR. Statistically significant differences in expression levels between time points are marked by * *p* < 0.05, *** *p* < 0.005.

cause non-infectious ("sterile") preterm labor. The latter is a model for 'idiopathic' human preterm labor, and one assumed to be caused by PR function loss [18] (Fig 2A). In mouse uteri infused with LPS, we assessed the expression levels of *Myb* and *Elf3* in preterm laboring myometrium compared to sham controls, or mice treated with a sterile saline infusion. We observed no significant differences in the levels of either target transcript between sham and LPS-treated mice (Fig 2B and 2C). Contrarily, upon examining the expression profiles of our genes of interest in the idiopathic preterm labor mouse model, we observed a different result: the transcript levels of both Myb and Elf3 were significantly elevated in the myometrium of RU486-treated mice relative to vehicle-treated mice (Fig 2B and 2C). These findings suggest that MYB and ELF3 may promote a laboring SMC state in both term and preterm labor, but not necessarily in response to an infection stimulus.

## The *Gja1* promoter is respectively activated and repressed in the presence of MYB and ELF3

Having established that *Myb* and *Elf3* are more highly expressed at the onset of labor relative to earlier gestational time points in select labor models, we hypothesized that these factors might coordinate contractility-driving gene expression. To address this hypothesis, we turned to an *in vitro* experiment system that had previously used luciferase reporter levels as a readout for transcription factor-mediated promoter regulation in a myometrial cell context. Prior studies had observed elevated reporter gene expression levels under the control of the *Gja1* promoter in Syrian hamster myometrial (SHM) cells in the presence of AP-1 heterodimers but not AP-1 homodimers [11, 12]. As previously outlined, *Gja1* is necessary for the initiation of contractions [9, 10] and the output of this gene promoter has often been used as a proxy for the laboring transcriptional program phenotype. We therefore generated a luciferase construct with a reporter gene under the control of the endogenous murine *Gja1* promoter to test the capacity of MYB and ELF3 in activating this promoter, either on their own or in the presence of AP-1 factors. Within the AP-1 FOS and JUN subfamilies, FOSL2 and JUND are especially highly expressed at the protein level at labor onset in mice [19], and we therefore selected these factors for our AP-1 homodimer and heterodimer test combinations. After first confirming the overexpression of transcription factor-encoding constructs introduced into SHM cells (**S2 Fig in** S1 File), we also verified that *Gja1* promoter activity increases with the addition of JUND, and is more significantly elevated in the presence of JUND and FOSL2 proteins, as previously reported [11, 12]. We then transfected the *Gja1* promoter-reporter gene plasmid and tested the activation levels of its promoter against a strong expression background of MYB or ELF3, with either factor alone or alongside AP-1 homodimers or heterodimers. First, we observed that the *Gja1* promoter was markedly activated in the presence of MYB alone compared to a transcription factor-absent background; furthermore, the addition of MYB to cells expressing JUND and FOSL2 resulted in a significantly increased reporter expression output compared to cells expressing only JUND and FOSL2 (Fig 3A). Having established that MYB can act as a mild activator of the *Gja1* promoter in myometrial cells, we next evaluated whether ELF3 might produce a similar result. Surprisingly, the reporter assay revealed that the overexpression of ELF3, either alone or in association with AP-1 dimers, resulted in levels of *Gja1* promoter activation that mirror those observed in the absence of any activating transcription factors (Fig 3B). Most strikingly, the addition of ELF3 in the JUND:FOSL2 background altogether eliminates the marked activation effect otherwise observed for the *Gja1* promoter in the presence of JUND and FOSL2. These findings suggest that, in myometrial cells, whereas MYB can act as an activator of *Gja1* promoter activity, ELF3 can contrarily act as a repressor in this context.

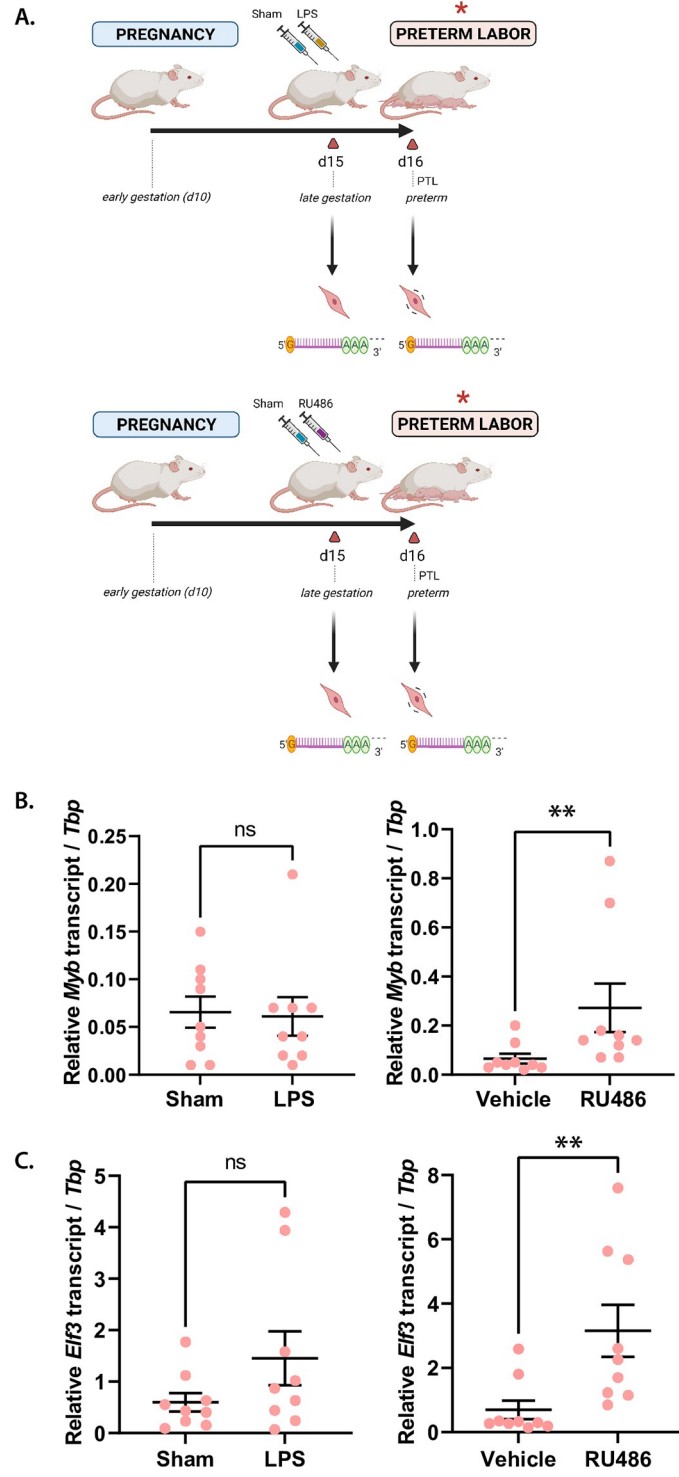

**Fig 2. Myb and Elf3 are up-regulated at labor onset in the RU486-, but not in the LPS-induced preterm labor model. (A)** Schematic of gestational time points of interest for RNA collection from CD-1 mouse myometrial tissues in two preterm labor models: local infection-simulating inflammation (LPS) and progesterone withdrawal (RU486). Gestational days (d) at which tissues were collected marked by triangles (red), with indication of injection with RU486 or LPS alongside vehicle and sham controls, respectively, at d15 and at preterm labor (18–24 hours post-infection (hpi), *italicized*). Expression levels (±S.D.) of transcription factor-encoding genes *Myb* **(B)** and *Elf3* **(C)** in sham- and LPS-treated mice (*left*) and in vehicle- and RU486-treated mice (*right*), as determined by RT-qPCR. Statistically significant differences in expression levels between control and preterm labor are marked by $^{**}$ $p < 0.01$.

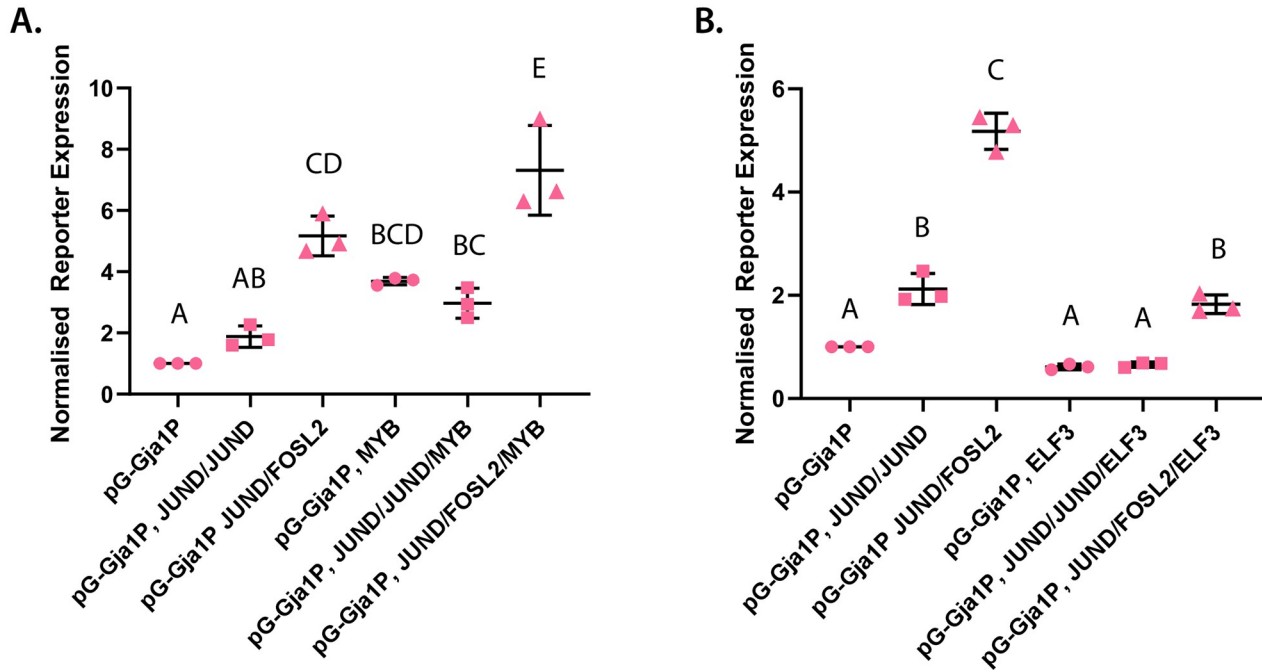

**Fig 3. MYB up-regulates while ELF3 down-regulates basal and AP-1 dimer-induced *Gja1* promoter activity.** Reporter expression levels from construct with luciferase reporter gene under control of murine *Gja1* promoter (Gja1P) relative to expression output of Renilla reporter plasmid. Cell treatment with **(A)** MYB alone, MYB and JUND:JUND homodimers, and MYB and JUND:FOSL2 heterodimers compared against control set in the absence of MYB; and **(B)** ELF3 alone, ELF3 and JUND:JUND homodimers, and ELF3 and JUND:FOSL2 heterodimers compared against control set in the absence of ELF3. Average expression level values (±S.D.) for each construct normalized to expression output of promoter construct subject to no transcription factor treatment. Groups determined by one-way ANOVA to be significantly different (p < 0.05) from one another are labeled with different letters; to indicate p > 0.05, groups are labeled with the same letter.

## AP-1 factors positively auto-regulate the *Fos* promoter in myometrial cells

Testing the effects of MYB and ELF3 on the *Gja1* promoter offer a preliminary indication of how these factors influence the progression of the labor program. Extending this functional analysis of these proteins to another promoter, however, affords the possibility of distinguishing potential gene promoter-specific effects of these candidate factors. Accordingly, we generated another reporter construct under the control of the endogenous murine *Fos* promoter. The genomic region upstream of this gene was selected in particular because of the previously observed up-regulation of *Fos* gene transcription in myometrial cells; the established role of the FOS protein in driving labor-associated gene promoter activation; and our previous findings that the *Fos* locus exhibits signs of elevated transcription during active murine labor relative to late pregnancy [8, 19–21]. Before testing the effects of either ELF3 or MYB on *Fos* promoter activity, however, unlike in the case of the *Gja1* promoter, the effects of JUN or FOS factors on this promoter had not yet been established in a myometrial context. We therefore first tested the effects of either JUND alone or FOSL2 and JUND together on reporter expression levels under the control of the *Fos* promoter in SHM cells. We observed a generally similar activation trend for the *Fos* promoter as in the case of the *Gja1* promoter: compared to treatment with no transcription factors, treatment of cells with JUND exhibited a slight elevation in reporter expression; the presence of JUND and FOSL2, however, significantly increased *Fos* promoter-mediated luciferase expression levels by a factor of over 5-fold (Fig 4A, ***left***). These

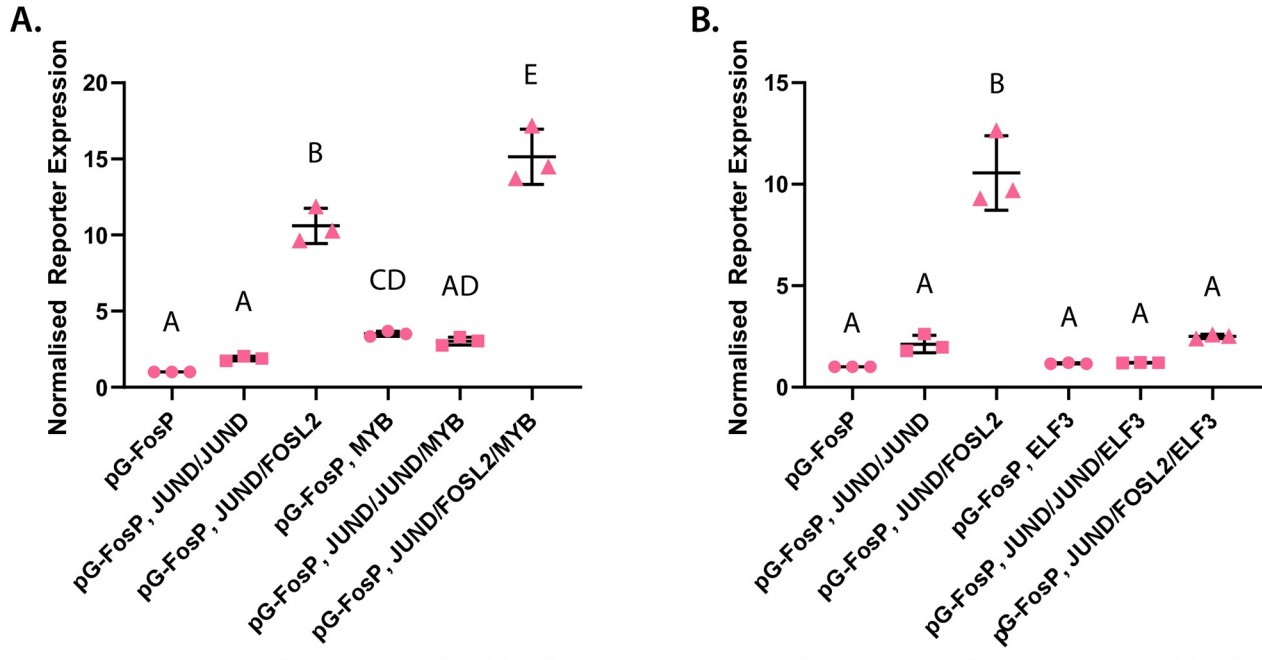

**Fig 4. MYB up-regulates while ELF3 down-regulates basal and AP-1 dimer-induced *Fos* promoter activity.** Reporter expression levels from construct with luciferase reporter gene under control of murine *Fos* promoter (FosP) relative to expression output of Renilla reporter plasmid. Cell treatment with **(A)** MYB alone, MYB and JUND:JUND homodimers, and MYB and JUND:FOSL2 heterodimers compared against control set in the absence of MYB; and **(B)** ELF3 alone, ELF3 and JUND:JUND homodimers, and ELF3 and JUND:FOSL2 heterodimers compared against control set in the absence of ELF3. Average expression level values (±S.D.) for each construct normalized to expression output of promoter construct subject to no transcription factor treatment. Groups determined by one-way ANOVA to be significantly different ($p < 0.05$) from one another are labeled with different letters; to indicate $p > 0.05$, groups are labeled with the same letter.

results suggest that the *Fos* gene may be auto-regulated by AP-1 factors in myometrial cells at the onset of labor.

## *Fos* promoter activities in the presence of MYB or ELF3 mirror the transcriptional effects observed at the *Gja1* promoter

Having established that AP-1 factors positively control *Fos* promoter activity in myometrial cells, we next evaluated the effects of MYB or ELF3, either individually or in the presence of AP-1 factors, on *Fos* promoter-mediated reporter expression levels. Here, again, *Fos* promoter activity followed the same regulatory trend as that observed in the case of the *Gja1* promoter: the addition of MYB resulted in a net upregulation of *Fos* promoter activity beyond levels observed in the absence of any transcription factors and in the presence of JUND and FOSL2 (Fig 4A). Similarly, the introduction of ELF3 on its own or alongside activating transcription factors caused a significant decrease in *Fos* promoter activity, significantly dampening any up-regulation effect otherwise observed in the presence of JUND and FOSL2 (Fig 4B). Considered together, these data collectively suggest that MYB is capable of activating labor-associated gene expression, both on its own and even further in the presence of FOS:JUN heterodimers. Furthermore, although Elf3 transcripts are more highly expressed at the onset of labor relative to late pregnancy, the ELF3 protein appears to be able to act as a repressor and even override the activation effects of AP-1 factors on labor-associated gene regulation.

## Discussion

Our findings collectively reflect the potential of MYB and ELF3 (albeit in apparently opposing roles) in modulating the gene expression program needed for SMCs to adopt a contractile state. We observed that both genes are more highly transcribed toward the end of the gestational period compared to one or more timepoints in the late gestation period in mouse and human models of term labor. We were surprised that, contrary to Bl6 mice and humans, CD-1 mice appeared to exhibit significantly increased *Myb* and *Elf3* transcript levels in SMCs by the term-not-in-labor stage, rather than at active labor. Because mouse strain-specific differences are unlikely to be the exclusive reason behind this discrepancy, further investigation would be warranted to account for these observations. Our further finding that *Myb* and *Elf3* undergo a particular spike in expression in the idiopathic preterm labor model associated with PR function loss, but not in the infection-simulating model, suggests that the corresponding proteins may be involved in particular pathways in laboring SMCs. MYB and ELF3 may be targeted specifically by the PR signaling pathway at term. Since AP-1 factor-mediated effects on *Gja1* promoter activity differ depending upon their PR isoform partner and its ligand status [12], other transcription factors may be likewise affected.

As our later results show, the AP-1 factor *trans*-activation potential can be altered in the presence of MYB or ELF3. The activation of *Gja1* and *Fos* promoter activity by MYB is consistent with evidence of this factor's function as a transcriptional activator in other studies, particularly in the context of cells from hematopoietic lineages [22]. Furthermore, an enrichment of MYB binding motifs in open chromatin genomic regions of pregnant human myometrium has previously been demonstrated (although this analysis excluded testing in an active laboring context) [1]. Thus, one possible and appealing explanation for the means by which MYB might impart the activation of labor-associated genes is through a direct transcriptional mechanism. Our observations lead us to postulate that MYB may be able to work in the same transcriptional complex with AP-1 heterodimers to boost the expression of labor-associated genes in term pregnant myometrium.

Compared to our findings regarding MYB protein activity, the role of ELF3 in contractile SMCs is less clear. As we demonstrated here, the *Elf3* gene expression trend from late pregnancy to labor onset is consistent with what was previously observed via microarray experiments, as in the case of *Myb* [23]. Indeed, we originally hypothesised that this transcription factor might play a role in up-regulating labor-associated genes, especially in light of previous demonstrations of ELF3-mediated induction of a matrix metalloproteinase gene promoter in cooperation with FOS:JUN heterodimers in chondrocyte cells [16]. Sinh et al. have also previously established a negative correlation in breast cancer cells between ELF3 levels and the expression of ZEB1 and ZEB2 [24], proteins that have been shown to repress the expression of the well-studied labor-associated genes, *Gja1* and *Oxtr* [25]. The converse repressive regulatory effect of ELF3 on *Gja1* and *Fos* promoter activities that we observed, however, does fall in line with certain findings related to ELF3-mediated gene regulation. ELF3 directly binds key labor program-driving ER alpha (ERα) and represses ERα-mediated gene transcription in breast cancer cells [26]; ELF3 has also been shown to decrease activation of labor-associated collagen genes in a non-SMC context [27]. These data collectively complement our finding that ELF3 plays a repressive role in labor-associated gene regulation.

The question remains as to why *Elf3* is highly transcribed at labor relative to day 15, a time point when its activity would assumedly be needed to support a labor-associated gene repression program. We present two separate, but not mutually exclusive hypotheses: since ELF3 has been shown in different cellular contexts and processes to act as an activator or a repressor, perhaps ELF3 can exert a gene-activating role during labor if the protein contains different

post-translational modifications or transcription factor partners that were not present in our current study. Examining the modification status of ELF3 at non-laboring and laboring stages would give a first indication if stage-specific differences exist that might impose opposing functional consequences. Alternatively, but not necessarily mutually exclusively, our finding might indicate a possible role for ELF3 in the early postpartum period, the onset of which requires a rapid reduction in labor-associated gene expression events. In this case, having high quantities of ELF3 readily available already during labor may assist in the successful transition from SMC contractility back to quiescence. Here, testing the effects of conditional blocking of *Elf3* expression before or at the labor onset stage in SMCs in mouse models can partially address this question by offering insights as to whether the postpartum involution process is consequently hindered or significantly altered.

Still further work needs to be done to uncover the exact molecular mechanisms by which MYB and ELF3 impart their activities in SMCs during labor-proximal gestational stages. Non-human cell lines and animal models offer a preliminary glance at the molecular players controlling contraction-driving genes. Establishing functional changes directly relevant to human parturition, however, requires additional experimental techniques. To test whether MYB and/ or ELF3 act as transcription factors in the same complex as AP-1 and PR factors, proof-of-interaction tests via co-immunoprecipitation or proximity ligation assay experiments in human tissues are needed. Chromatin immunoprecipitation experiments can also indicate whether either protein directly binds open chromatin regions in or near contraction-associated genes in human term-not-in-labor or active laboring SMCs. More broadly, If MYB and ELF3 play an essential role in establishing the contractile physiological phenotype in the myometrium, uterine contractility tests on transcription factor overexpression models like those developed by Peavey et al. for PRs [28] can best address this question.

As they stand, our findings are important to consider in light of potential therapeutic applications that can be developed to address the critical health challenges associated with early labor onset. For instance, as we previously discussed [13], attempts to target MYB with different drugs or compounds to downregulate its activity in the context of treating acute myeloid leukemia have been met with some success [29–31]. Of course, therapeutic treatment of human extreme preterm birth may mandate somewhat different strategies than that of cancer. Nevertheless, much recent work shows promise in the targeting of genomic activity in SMCs in order to prolong pregnancy to term. For instance, our earlier study uncovered gestational time point-specific, genome-wide active histone mark patterns; the observation of these patterns support previous findings that the onset of labor is associated with global histone deacetylation [8, 32, 33]. Recently, Zierden and others have developed a mode of delivery of suspensions containing inhibitors of histone deacetylase proteins alongside the labor-stalling hormone, progesterone; their results showed that these two-target suspensions provide greater efficacy in preterm birth prevention in bacterial infection-induced mouse labor models than do two prominent approved drugs [34]. In light of these findings, and with further inquiry into the regulation mechanisms of MYB and ELF3, the most effective therapeutic measures toward preterm birth might involve pinpointing multiple targets, including these two proteins.

## Materials and methods

### Ethics statement

This study was carried out in accordance with the protocol approved by the Research Ethics Board, Sinai Health System REB# 02-0061A and REB# 18-0168A. All subjects donating myometrial biopsies for research gave written informed consent in accordance with the Declaration of Helsinki. All research using human tissues was performed in a class II-certified

laboratory by qualified staff trained in biological and chemical safety protocols, and in accordance with Health Canada guidelines and regulations.

All mouse experiments were approved by the Animal Care Committee of The Centre for Phenogenomics (TCP) (Animal Use Protocol #0164H). Guidelines set by the Canadian Council for Animal Care were strictly followed for handling of mice. Virgin outbred CD-1 or inbred Bl6 (C57/Bl6) mice used in these experiments were purchased from Harlan Laboratories (http://www.harlan.com/).

## Human myometrial tissue collection

After receiving written consent, myometrial biopsies from healthy women undergoing term-not-in-labor elective Caesarean sections (TNL, n = 5) or spontaneous, non-induced term labor (TL, n = 5) were collected and transferred from the operating theatre to the laboratory. For the TL group, biopsies were obtained during emergency Caesarean sections due to breech presentation and fetal distress. Patients with preterm premature rupture of membranes (PPROM), clinical chorioamnionitis, fetal anomalies, gestational diabetes/hypertension, cervical cerclage, preeclampsia, antepartum hemorrhage, and autoimmune disorders were excluded from the study. After the delivery of the fetus and placenta, a small 1.5 cm sliver of myometrium was collected from the upper margin of the incision made in the lower uterine segment in both laboring and non-laboring women prior to closing of the uterus. The endometrium was removed, and myometrial tissues were flash-frozen in liquid nitrogen and stored at −80˚C until needed.

## Mouse gestational model tissue collection

All animals were housed in a pathogen-free, humidity-controlled 12h light, 12h dark cycle TCP facility with free access to food and water. Female CD-1 and Bl6 mice were naturally bred; the morning of vaginal plug detection was designated as gestational day (d) 1. Pregnant mice were maintained until the appropriate gestational time point. For term labor models, gestation in mice on average is up to 3 weeks. Delivery under these conditions occurred during the evening of d19 or the morning of d20. Our criteria for labor were based on delivery of at least one pup.

In the term labor model, samples were collected on gestational days 15, 17, 19 term-not-in-labor (TNIL), 19–20 during active labor (TL), and 2–8 hours postpartum (PP) for CD-1 mice and on gestational days 15, 18.75 (TNIL), 19–20 during active labor (TL) and 2–8 hours PP for Bl6 mice. Myometrial tissues were collected at 10 AM on all gestational days with the exception of labor samples (TL), which were collected once the dams had delivered at least one pup. In the infection-associated inflammation preterm labor model, mice underwent a mini-laparotomy under isoflurane for anesthesia on gestational day 15, and were given an intrauterine injection of either sterile saline (sham) or 125 µg of *E. coli*-derived LPS (serotype 055:B5) in sterile saline. Mice were sacrificed 24 hours after sham surgery or during LPS-induced preterm labor, which occurred 24 hours post infection +/- 6 hours. In the idiopathic preterm labor model, mice were subcutaneously injected with vehicle (corn oil/ethanol) or 150 µg of mifepristone (RU486) on gestational day 15. Mice were sacrificed 24 hours after injection or during RU486-induced preterm labor, which occurred 24 hours post injection +/- 2 hours.

Mice were euthanized by carbon dioxide inhalation. The part of the uterine horn close to the cervix from which the fetus was already expelled during term or preterm labor was removed and discarded; the remainder was collected for analysis. For each day of gestation, tissue was collected from 4–8 different animals. Isolated uteri were placed into ice-cold PBS. Uterine horns were bisected longitudinally and dissected away from both pups and placentas. The decidua basalis was cut away from the myometrial tissue. The decidua parietalis was

carefully removed from the myometrial tissue by mechanical scraping on ice, which eliminated the entire luminal and glandular epithelium and most of the uterine stroma. Myometrial tissues were flash-frozen in liquid nitrogen and stored at −80˚C until needed.

## Gene expression quantification by RNA extraction and RT-qPCR

Myometrial tissues were crushed into a fine powder on dry ice. Total RNA was extracted using Trizol and further treated with DNase I to remove genomic DNA. RNA was reverse transcribed using the high-capacity cDNA synthesis kit (Thermo Fisher Scientific). Target gene expression was monitored by qPCR using SYBR Select (Thermo Fisher Scientific) and primers that target sequences within the exons of pertinent mRNA transcripts (**S1, S2 Tables in** S1 File), and normalized to levels of total *H1f0* (Bl6 mice), *Tbp* (CD-1 mice) or *Mapk1* (human) mRNA. These reference genes have been previously used because of their most consistent profiles of expression at similar levels across gestational time points [8, 19]. Relative expression levels were calculated against genomic DNA-based standard curve references. All samples were confirmed not to have DNA contamination because no target amplification was observed with DNA polymerase in the qPCR reactions for reverse transcriptase-negative samples.

## Syrian hamster myometrial (SHM) cell culture

Syrian hamster myometrial (SHM) cells supplied by Professor Oksana Shynlova were cultured in phenol-free DMEM supplemented with 10% fetal bovine serum, 100 IU/ml penicillin, and 100 µg/ml streptomycin (Pen-Strep). Cells were kept in a 37˚C/5% $CO_2$ environment and passaged and/or had their media replaced, as appropriate, every 2–4 days. Cell media was tested and confirmed negative for presence of mycoplasma via Mycoplasma PCR Detection Kit (Biovision, catalog #K1476-100).

## Reporter plasmid cloning and acquisition

For luciferase assays, we utilized pGL4.23 (luciferase vector) or modified versions thereof and pGL4.23 (Renilla vector) from Promega. To generate luciferase constructs under the control of *Gja1* and *Fos* promoters (Addgene catalog #188114 and #188113, respectively), the promoters were amplified from Bl6 murine genomic DNA. The original minimal promoter from the pGL4.23 construct was first removed via digestion with BglII and NcoI and purification of the plasmid backbone. The labor-associated endogenous gene promoters were cloned into the pGL4.23 backbone (Promega) either directly from gDNA (using overhang-containing primers) in the case of the *Fos* promoter, or from transitional pJET-1.2 vector backbones containing the cloned promoter in the case of the *Gja1* promoter (**S3 Table in** S1 File). All transcription factor-encoding constructs consisted of a pcDNA3.1 plasmid backbone, with the exception of the plasmid encoding ELF3, which consisted of a pCI plasmid backbone. The potential regulatory activity of any one transcription factor-encoding plasmid was always compared to its gene-free plasmid backbone counterpart. FOSL2- and JUND-encoding plasmids are described in Mitchell and Lye [11] and can be obtained from Addgene (catalog #187907 and #187904, respectively). The pCl and pCl-ELF3 constructs [35] were a kind gift from Dr. Miguel Otero and Dr. Mary B. Goldring, Hospital for Special Surgery, New York. The MYB-FLAG construct was obtained from Addgene (catalog #66980 [36]).

## Western blot

SHM cells were transfected with 100 ng of construct/well of 4 wells in 24-well plate format 24h post-seeding. 48h post transfection, media was removed and cells were washed with PBS

and frozen at -80˚C. Cells were lysed in York lysis buffer (1M Tris/HCl pH 6.8, 10% SDS (final conc. 2%), glycerol (final conc. 10%), and ddH$_2$O, with proteinase and phosphatase inhibitor cocktail (Halt™ Protease and Phosphatase Inhibitor Cocktail, EDTA-free, Thermo Scientific). Cells were scraped out of wells, with pooling of well contents from 4 wells pertaining to the same transfected vector on ice. Samples were kept cool as debris was removed and sonicated prior to denaturing of proteins by heating at 95˚C and to leaving samples to cool on ice for 5 min each. Following a final centrifuge spin, lysate-containing supernatants were collected and protein quantities were determined via BCA (Pierce™ BCA Protein Assay Kit, Thermo Scientific). Proteins were denatured by addition of 2X Laemmeli buffer/10% β-mercaptoethanol (Bio-Rad/Sigma, respectively) to samples and subjection of samples to boiling for 2 min at 95˚C. Samples were incubated on ice, centrifuged at 4˚C, and supernatants were transferred to new tubes. 20 μg of protein lysate from each sample were loaded into each lane of a 10% SDS-PAGE gel. Contents were separated prior to transfer onto a nitrocellulose membrane. Membranes were blocked by incubation with blocking solution (2.5% milk) prior to addition of the appropriate antibody in 2.5% or 5% milk solution (1/1000 dilution anti-JUND, sc-74 (Santa Cruz); 1/1000 dilution anti-FOSL2, sc-171 (Santa Cruz); 1/2000 dilution anti-ELF3, A6371 (ABclonal); 1/2000 dilution anti-FLAG, F1804 (Millipore Sigma)) for incubation overnight at 4˚C. Membranes were washed three times in TBST prior to incubation with secondary antibody (1/5000 dilution horseradish peroxidase (HRP)-conjugated anti-mouse (Bio-Rad, 170–6516) or anti-rabbit (Bio-Rad, 170–6515)), as appropriate) for 30 min. Membranes were washed three times in TBST prior to incubation in ECL solution (Western Lightning Plus, Chemiluminescent Substrate, PerkinElmer) for 5 min. Membranes were developed and images were visualized (**S2 Fig in** S1 File) using Bio-Rad Image Lab Software.

## Luciferase assays

Activity of predicted labor-associated transcription factor candidates was assayed using a dual luciferase reporter assay (Promega). On Day 0, SHM cells were seeded in antibiotic-free media at a density of $5x10^4$ cells/well within a 24-well plate format. On Day 1, media was replaced with Opti-Mem 1hr prior to transfection. Using Lipofectamine 3000, molar equivalents of the appropriate transcription factor-encoding constructs were transfected into cells, alongside a 1:1 molar ratio of Luciferase (pGL4.23) to Renilla (pGL4.75, Promega) vectors, which we determined as the optimised ratio for most uniform Renilla levels across vector treatments in these assays. Total transfected DNA for any one vector combination set did not exceed a maximum amount of 500 ng/well. On Day 2, Opti-Mem was replaced with fresh antibiotic-free SHM media. On Day 3, media was removed and cells were lysed in 1X passive lysis buffer/ PBS, incubated at RT for 15 min, and frozen at -80˚C for at least one hour and up to one week. Reporter activity was measured on the Fluoroskan Ascent FL plate reader 48 hpt and calculated by normalising firefly expression to *Renilla* expression.

## Statistical analyses

Significant changes in myometrial gene expression in mouse term gestational profiles were determined by one-way ANOVA with Tukey correction. Changes in gene expression in mouse preterm labor models were determined by unpaired t-tests using GraphPad Prism 9. Human myometrial gene expression data were analyzed by unpaired t-tests using GraphPad Prism 9. Luciferase assay data were analyzed by two-way ANOVA using Sigma Plot12 and significant differences were confirmed by the Holm-Sidak method.

## Supporting information

**S1 File. Supporting Information for: MYB and ELF3 differentially modulate labor-inducing gene expression in myometrial cells.**
(DOCX)

**S1 Raw images. Raw images for S2 Fig in S1 File.**
(PDF)

## Acknowledgments

We would like to thank Luis Abatti for his advice pertaining to construct design, and Dr. Lubna Nadeem for her extensive guidance and instruction in the SHM cell culture and luciferase assay experiment design. We also thank the human myometrial specimen donors, the Research Centre for Women's and Infants' Health BioBank at the Lunenfeld-Tanenbaum Research Institute, and the Department of Obstetrics and Gynecology at Sinai Health System, particularly the staff in the Labor and Delivery Unit for their support in the human specimen collection used in this study. Select figures in this study were created using BioRender (Biorender.com).

## Author Contributions

**Conceptualization:** Virlana M. Shchuka, Jennifer A. Mitchell.

**Data curation:** Virlana M. Shchuka.

**Formal analysis:** Virlana M. Shchuka, Nawrah Khader.

**Funding acquisition:** Virlana M. Shchuka, Oksana Shynlova, Jennifer A. Mitchell.

**Investigation:** Virlana M. Shchuka, Nawrah Khader.

**Methodology:** Virlana M. Shchuka, Nawrah Khader, Anna Dorogin.

**Project administration:** Virlana M. Shchuka, Oksana Shynlova, Jennifer A. Mitchell.

**Resources:** Oksana Shynlova, Jennifer A. Mitchell.

**Supervision:** Jennifer A. Mitchell.

**Validation:** Virlana M. Shchuka, Nawrah Khader.

**Visualization:** Virlana M. Shchuka, Nawrah Khader.

**Writing – original draft:** Virlana M. Shchuka.

**Writing – review & editing:** Virlana M. Shchuka, Nawrah Khader, Anna Dorogin, Oksana Shynlova, Jennifer A. Mitchell.

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
