## [Decision Letter · Decision Letter 0]

28 Sep 2022

PONE-D-22-17810MYB and ELF3 differentially modulate labor-inducing gene expression in myometrial cellsPLOS ONE

Dear Dr. Mitchell,

Thank you for submitting your manuscript to PLOS ONE. After careful consideration, we feel that it has merit but does not fully meet PLOS ONE’s publication criteria as it currently stands. Therefore, we invite you to submit a revised version of the manuscript that addresses the points raised during the review process.

We look forward to receiving your revised manuscript.

Kind regards,

Atsushi Asakura, Ph.D

Academic Editor

PLOS ONE

Journal Requirements:

2. We note that Figure (1 and 2) in your submission contain copyrighted images. All PLOS content is published under the Creative Commons Attribution License (CC BY 4.0), which means that the manuscript, images, and Supporting Information files will be freely available online, and any third party is permitted to access, download, copy, distribute, and use these materials in any way, even commercially, with proper attribution. For more information, see our copyright guidelines: http://journals.plos.org/plosone/s/licenses-and-copyright.

1. You may seek permission from the original copyright holder of Figure (1 and 2) to publish the content specifically under the CC BY 4.0 license. 

Reviewers' comments:

Reviewer's Responses to Questions

**Comments to the Author**

1. Is the manuscript technically sound, and do the data support the conclusions?

Reviewer #1: Yes

Reviewer #2: Yes

2. Has the statistical analysis been performed appropriately and rigorously? 

Reviewer #1: Yes

Reviewer #2: I Don't Know

3. Have the authors made all data underlying the findings in their manuscript fully available?

Reviewer #1: Yes

Reviewer #2: Yes

4. Is the manuscript presented in an intelligible fashion and written in standard English?

Reviewer #1: Yes

Reviewer #2: Yes

5. Review Comments to the Author

Reviewer #1: Title: MYB and ELF3 differentially modulate labor-inducing gene expression in myometrial cells

In this study, MYB and ELF3 gene expressions were assayed in the human and murine myometrium of different gestational ages including term not in labor and in labor. Next, the authors utilized two different preterm mouse models: infection-caused preterm model (LPS injection) and progesterone withdrawal preterm model (RU486 injection). Both Myb and Elf3 mRNA levels were increased by RU486 injection, but not by LPS. Using a sophisticated luciferase construct with a reporter gene under the control of the Gja1 promoter to test the capacity of MYB and ELF3 in activating this promoter, the authors showed that Gja1 and Fos promoter activation increased in the presence of MYB, but rather decreased in the presence of ELF3. As the authors mentioned, MYB can act as an activator of Gja1 promoter activity, whereas ELF3 can act as a repressor in myometrial cells. Finally, the authors tested the auto-regulation (feed-forward fashion) of the Fos promoter in myometrial cells. MYB upregulated Fos promoter activity even in the absence of JUND and FOSL2, and MYB further upregulated it in the presence of JUND and FOSL2. In contrast, ELF3 suppressed the Fos promoter activity in the presence of JUND and FOSL2.

Comments:

1) Lines 83-86. Introduction section: The authors picked up two genes, MYB and ELF3 but the reasons are unclear. Please explain why these two genes were focused in this study.

2) Lines 83-86. The explanations of MYB ELF3 are necessary in the Introduction section because these genes are not familiar or famous in the field of reproduction. Please explain the functions of MYB and ELF3 in other tissues citing the previous published papers.

3) Lines 83-86. What is the functional relationship between MYB and ELF3? This is a very important point in this paper but it is not mentioned throughout the manuscript. These two genes were analyzed in parallel, but if these two molecules were not interacting with each other, it is nonsense to show the data of MYB and ELF3 together.

4) Figure. 1. In mice, both Myb and Elf3 mRNA levels have been already increased at D19 term not in labor, and the expression levels were as high as those in labor. However, in humans, the gene expressions of MYB and ELF3 were lower in term not in labor than those of term in labor. What causes the difference of increase pattern between mice and humans?

In sFig. 1, the Myb and Elf3 gene expression patterns of Bl6 mice are similar to those of humans, not to CD-1 mice. It is likely that there is a difference in the expression pattern between mice strains. Please discuss it.

5) Lines 166-168. The information that the elevated reporter gene expression levels under the control of the Gja1 in the presence of AP-1 heterodimer but not AP-1 homodimers (ref. 11,12) is very important to understand the results of Fig. 3. Please introduce the Nature Communication paper (Ref 12) in the Introduction section.

6) Line 169. Brief explanation of Gja1 gene is warranted here. This gene corresponds to CX43, a gap junction protein.

7) Line 173. The brief explanation of JUND and FOSL2 is also necessary here or in the Introduction section. I know that these genes are the components of Jun and Fos subfamily, respectively, but the AP-1 transcriptional factors and its related molecules should be concisely explained for readers.

8) The data in Figure 3 are very interesting. Gja1 and Fos promoter activation increased in the presence of MYB. In contrast, ELF3 suppressed the Gja1 and Fos promoter activation. This luciferase assay-construct is elegant. This is just a comment.

9) The paragraphs of “AP-1 factors positively auto-regulate the Fos promoter in myometrial cells” (lines 200-221) and “Fos promoter activities in the presence of MYB or ELF3 mirror the transcriptional effects observed at the Gja1 promoter” (lines 233-249) are the result of same figure, Figure 4, so it might be better to combine these two paragraphs together.

10) Overall, I recommend describing the key molecules (for example, Gja1, AP-1, Jun, Fos, etc) in the Introduction section. Most of these molecules are introduced in the Result section at present, so it is hard to follow for readers.

Reviewer #2: SUMMARY:

The authors examined two genes, Myb and Elf3, identified to be up-regulated during mice labor in a previously published study that utilised total RNA-seq. Firstly, they correlated the myometrial tissue expression levels of their transcripts to (i) different stages of in vivo physiological pregnancy and labor in mice (d15, d17, d19, d19.5 and d20; all five time points for CD-1 and four for Bl6 strain) and humans (no labor and labor; both term gestation), and (ii) two established in vivo mouse models of preterm labor (LPS and RU486 exposure). Secondly, luciferase-based reporter gene assays were used to demonstrate the effects of Myb and Elf3 expression, alongside genes that encode FOSL2 and JUND (AP-1 monomers), from plasmids transfected into immortalised SHM cells on the activities of (iii) Gja1 and (iv) Fos gene promoters (cloned from Bl6 genomic DNA). Results show (i) previous RNA-seq findings validated using qPCR, (ii) RU486 increases Myb and Elf3 transcripts more so than LPS, and (iii)-(iv) Myb expression enhances both Gja1 and Fos gene promoter activity but the opposite was observed from Elf3 expression. Together, these preliminary findings present Myb and Elf3 as novel labor-associated transcription factors of interest.

SPECIFIC COMMENTS:

Abstract

-Minor:

• Lines 25-27: the first letter of each transcription factor (full) name should be lower case.

• Line 34: use lower case for ‘lipopolysaccharide’.

Introduction

Concise and well-structured; relevant references have been included.

-Major:

• Lines 83-86: it would be helpful to briefly explain why MYB and ELF3 were chosen as novel proteins of interest in the Introduction, rather than this first being provided in the Results (lines 104-105), section. Furthermore, explicit hypotheses/aims of the study should be stated to justify the choices of experiments that were undertaken.

-Minor:

• Line 46: please define “term” gestation for PLOS readers who may not be familiar with obstetrics terminology.

Results

-Major:

• Line 138-139: it should be explicitly stated that the modelling of ‘idiopathic’ human preterm labor using mice in the present study is based on the assumption (not definitively proven) that human idiopathic preterm labor is caused solely by loss of progesterone receptor function when using RU486.

• All figures: please state in the captions what the error bars are representative of (e.g. mean ± SEM or SD).

• Fig S2: can the authors provide data to show MYB protein abundance in transfected cells (alongside the FLAG tag abundance data shown)?

• Fig 3 and 4: captions for these figures state “Average expression level values for each construct normalised to expression output of promoter construct subject to no TF treatment” – does this mean that cells transfected with both TF construct(s) and pG-Gja1P were all normalised to cells with only pG-Gja1P, and hence why there are no error bars for the ‘pG-Gja1P’ group? If all values from cells transfected with TF constructs were normalised to ‘no TF treatment’, does this mean that ‘no TF treatment’ values used for ANOVA were all ‘1’? Why not present (and undertake statistical analysis on) the values “calculated by normalising the firefly to Renilla expression” (line 437) without further normalisation to the ‘pG-Gja1P’ group?

• Fig 3 and 4: please state why JUND and FOSL2 were the chosen members of the AP-1 family of proteins to examine using the reporter assay, and thus why the other AP-1 monomers appear to not be relevant to this study.

-Minor:

• Fig 1, 3, 4, S1: is it possible for the authors to add or use alternative annotations to indicate statistical significance on graphs for ANOVA? For example, in Fig 3, it is unclear whether ‘pG-Gja1P, JUND/JUND’ vs ‘pG-Gja1P, JUND/JUND/MYB’ are significantly different according to ANOVA.

• Fig 2: check style consistency (symbol colours and error bar thickness) of ‘B’ graph on right.

• Fig 2 caption (line 154): please give the full name for “hpi” if only used once in the manuscript; also check that the ‘B’ and ‘C’ labels for the graphs match the description in the caption.

• Fig S2: please state full names for all abbreviations used to label the figure in the caption.

Discussion

Good consideration of recent developments relevant to MYB and ELF3.

-Major:

• Please add a ‘Study limitations’ paragraph to briefly discuss, for example, (i) limits of using cell lines and animal models to infer expression/function-related changes during human parturition (particularly for addressing the discrepancy between in vivo gene expression and in vitro gene promoter activity data for Elf3), (ii) absence of molecular biology techniques that assess interactions between transcription factors as postulated by the authors (e.g. line 262-265), and (iii) absence of functional assays beyond transcription factor/gene promoter activity that would truly demonstrate the impact of MYB and ELF3 expression on myometrial phenotype (amongst all other changes that will occur at a multitude of other proteins during physiological and pathological labor).

• Lines 282-290: it would be good for the authors to present suggestions for experiments to be done to address the two conjectures proposed to explain high Elf3 expression in vivo despite repressive effects on gene promotor activity in vitro (to elaborate on “Further work” at line 291).

• Please elaborate on the difference observed for Myb and Elf3 expression between LPS and RU486 mice preterm labor models, and explicitly state the relevance of AP-1 to the RU486 model that would further justify it being the focus of the reporter assays.

Materials & Methods

-Major:

• A good description of how mice myometrium tissues were dissected and stored after removal from the body has been included (lines 362-367). Please add an equivalent description for human myometrium tissues in the ‘Human tissue collection’ section.

• Lines 325-327: were cases of artificial induction and augmentation of labor also excluded?

• Lines 375-376: either include Supplemental Data and/or citations to support the statement that “these reference genes were most consistently expressed at similar levels across gestational time points”.

• Line 377-379: review the technical accuracy and/or whether the positioning of this statement at the end of the paragraph makes sense. Were the authors referring to checking for genomic DNA contamination during qPCR, which uses DNA polymerase, or at conversion of RNA to cDNA using reverse transcriptase (and checked using agarose gel electrophoresis)?

• “Reporter plasmid cloning and acquisition” section (pages 18-19): insert mention of S2 Fig, and also state whether transfection efficiency for cells used for reporter assays was checked for all n=3 shown in Fig 3 and 4.

• Table S3: check all primer sequences provided are accurate, as well as their text formatting.

• Line 423: please state what equipment (and software) was used to acquire the (digital) images of the ECL reagent-treated membranes, as well as briefly describe how the images were analysed.

Minor:

• Details for animal research ethics (lines 332-336) would be better placed under the same ‘Ethics statement’ heading (page 15) used for the ethics described for the human-derived samples, but as a separate paragraph. The remainder of the ‘Animal model’ section (lines 336-342) could then form the first paragraph of the ‘Mouse gestational model myometrial tissue collection’ headed section (page 16) instead.

• Line 320: please add “Human” to the start of the ‘tissue collection’ heading to better distinguish from the section that describes mice tissue collection.

• Line 324: do the authors mean ‘breech’ where it states “bridge” presentation?

• Would it be possible for the authors to present median with range values for gestation age, gravida, and parity for their human study participants – potentially in a Supplemental Table? BMI, ethnicity, whether participants had a previous caesarean section, and presence/absence of fetal membrane rupture immediately prior to caesarean section would also be of interest if available.

• Line 350: insert ‘preterm’ before “labor”.

• Line 373: state what qPCR reagent was used with the cDNA samples and primers (e.g. SYBR Green?).

• Lines 374-375: state which housekeeping gene was assigned to each mouse strain (as indicated in Fig 1 and S1).

• Tables S1, S2 & S3: please add RefSeq accession numbers used to confirm specificity of each pair of primers (using e.g. NCBI Primer-BLAST).

• Lines 382-383: Abbreviations – “DMEM media” technically uses the word ‘media’ twice; same applies to “Opti-MEM media” at lines 433-434. Please state the full name for “FBS” (instead of using the abbreviation if only used once in the manuscript).

• Lines 381-385: please state where the SHM cells were sourced from. Purchased commercially or donated by a named collaborator?

• Line 382: specify type of plate if “plate” is to be used in this sentence.

• Line 383: state the working concentrations of penicillin and streptomycin in the media used instead of stating “1%”.

• Line 385: please state what method of mycoplasma testing was used.

• Line 395: what was the “pCI” construct used for?

• Lines 395-397: check the relevance of references 29 and 30 for their corresponding sentences.

• Lines 402 and 435: missing degrees symbol in “-80C”.

• Line 417: please check the catalog number or supplier info for anti-ELF3.

Other minor points

• Check “Hist1” is the correct gene name (and not referring to a gene cluster).

• ‘MAPK1’ is the official gene name for ERK2; the latter is considered a synonym.

• Check that Greek letters are used instead of their Latin substitutes (e.g. “u” (where it should be ‘�’) at lines 356 and 412).

• Please be consistent with use of abbreviations. For example, “GD” was used to abbreviate ‘gestation day’ in line 339, but then “gestation day” was used throughout the following ‘Mouse gestational model myometrial tissue collection’ section in Materials and Methods instead of ‘GD’. Same with “pp” for ‘postpartum in lines 347-348.

• Please provide full name for the “TF” abbreviation at its first use (line 178?).

• Check citations. For example, according to the References list, Sinh is first author for reference 20 (not 19) in lines 272-274, and cited reference 28 in line 307 looks like it should be 29.

6. PLOS authors have the option to publish the peer review history of their article (what does this mean?). If published, this will include your full peer review and any attached files.

Reviewer #1: No

Reviewer #2: No

---

## [Author Response · Author response to Decision Letter 0]

15 Nov 2022

NOTE: in our response all references to page and line numbers refer to the version with changes tracked.

RESPONSE to REVIEWER 1

Reviewer #1: Title: MYB and ELF3 differentially modulate labor-inducing gene expression in myometrial cells

In this study, MYB and ELF3 gene expressions were assayed in the human and murine myometrium of different gestational ages including term not in labor and in labor. Next, the authors utilized two different preterm mouse models: infection-caused preterm model (LPS injection) and progesterone withdrawal preterm model (RU486 injection). Both Myb and Elf3 mRNA levels were increased by RU486 injection, but not by LPS. Using a sophisticated luciferase construct with a reporter gene under the control of the Gja1 promoter to test the capacity of MYB and ELF3 in activating this promoter, the authors showed that Gja1 and Fos promoter activation increased in the presence of MYB, but rather decreased in the presence of ELF3. As the authors mentioned, MYB can act as an activator of Gja1 promoter activity, whereas ELF3 can act as a repressor in myometrial cells. Finally, the authors tested the auto-regulation (feed-forward fashion) of the Fos promoter in myometrial cells. MYB upregulated Fos promoter activity even in the absence of JUND and FOSL2, and MYB further upregulated it in the presence of JUND and FOSL2. In contrast, ELF3 suppressed the Fos promoter activity in the presence of JUND and FOSL2.

We would like to thank the reviewer for their assessment of our work.

Comments:

1) Lines 83-86. Introduction section: The authors picked up two genes, MYB and ELF3 but the reasons are unclear. Please explain why these two genes were focused in this study.

As requested, we have provided our justification for focusing on MYB and ELF3 in the laboring myometrial context in the introduction section. On pgs. 5-6, lines 94-109, the text now reads:

“We hypothesized that these two proteins might act as potential regulators of the laboring SMC phenotype for several reasons: both their corresponding genes are more highly transcribed at term relative to earlier quiescent gestational stages in mouse myometrial tissues [7,8,14]. Associated predicted binding motifs for both proteins are also contained within promoters of contraction-associated genes like Gja1. More notably, MYB and ELF3 have been shown to behave as transcriptional activators in cooperation with AP-1 factors in other cellular contexts. For example, Tapias and others determined that, in HeLa cells, MYB both controls the transcription of and co-localizes with AP-1 factors at the promoter of Sp3 [15], a gene prominently expressed in SMCs [reviewed in 13]. Likewise, a study by Otero and others demonstrated that ELF3 binds the Mmp13 gene promoter in chondrocyte cells; while AP-1 heterodimers can activate the Mmp13 promoter, the addition of ELF3 further compounds this activation effect [16]. ELF3 has also been shown to interact with Nf-κβ in prostate cancer cells in response to pro-inflammatory stimuli that affect late-gestation SMCs [17]. In light of these findings, we hypothesized that MYB and ELF3 might be significant members of the laboring SMC transcription regulatory network, and ones that function in conjunction with AP-1 factors.”

2) Lines 83-86. The explanations of MYB ELF3 are necessary in the Introduction section because these genes are not familiar or famous in the field of reproduction. Please explain the functions of MYB and ELF3 in other tissues citing the previous published papers.

As stated in our response to comment 1), this explanation is now provided on pgs. 5-6, in lines 94-109. Briefly, we summarized the findings from several critical studies regarding these proteins’ functions as activating transcription factors in different cell types, which included determining the binding partners and target gene promoters of MYB and/or ELF3 in HeLa cells (Tapias et al., 2008), chondrocyte cells (Otero et al., 2012), and prostate cancer cells (Longoni et al., 2013). 

3) Lines 83-86. What is the functional relationship between MYB and ELF3? This is a very important point in this paper but it is not mentioned throughout the manuscript. These two genes were analyzed in parallel, but if these two molecules were not interacting with each other, it is nonsense to show the data of MYB and ELF3 together.

As stated in our response to comment 1), the justification for focusing on MYB and ELF3 together is now provided on pgs. 5-6, in lines 94-109. Briefly, since MYB and ELF3 are upregulated during labor in SMCs and appear, in other cellular contexts, to interact with the same binding partners (AP-1 factors) to exert an activation effect on target gene families whose expression is critical for SMCs in late gestation and/or during active labor, we originally hypothesized that these two factors might both be involved in AP-1-mediated gene regulation, and we therefore focused on both these factors in this paper.

4) Figure. 1. In mice, both Myb and Elf3 mRNA levels have been already increased at D19 term not in labor, and the expression levels were as high as those in labor. However, in humans, the gene expressions of MYB and ELF3 were lower in term not in labor than those of term in labor. What causes the difference of increase pattern between mice and humans?

In sFig. 1, the Myb and Elf3 gene expression patterns of Bl6 mice are similar to those of humans, not to CD-1 mice. It is likely that there is a difference in the expression pattern between mice strains. Please discuss it.

We were surprised by this finding and have acknowledged this discrepancy in a new paragraph in the discussion section on pgs. 14-15, lines 298-303. The excerpt pertaining to this comment now reads:

“We were surprised that, contrary to Bl6 mice and humans, CD-1 mice appeared to exhibit significantly increased Myb and Elf3 transcript levels in SMCs by the term not in labor stage, rather than at active labor. Because mouse strain-specific differences are unlikely to be the exclusive reason to account for this discrepancy, further investigation would be warranted to account for these observations.”

5) Lines 166-168. The information that the elevated reporter gene expression levels under the control of the Gja1 in the presence of AP-1 heterodimer but not AP-1 homodimers (ref. 11,12) is very important to understand the results of Fig. 3. Please introduce the Nature Communication paper (Ref 12) in the Introduction section.

As requested, we have provided a succinct overview of the central findings regarding AP-1-mediated control of Gja1 promoter activity from the referred-to studies by Mitchell and Lye (2005) and Nadeem et al. (2016) in the introduction section, on pg. 4, in lines 78-80. The text now reads:

“For example, AP-1 heterodimers, but not homodimers, tend to activate the murine Gja1 promoter [11], and this heterodimeric activation effect can be heightened or hampered depending on which major PR isoform is present in SMCs [12].”

6) Line 169. Brief explanation of Gja1 gene is warranted here. This gene corresponds to CX43, a gap junction protein.

As requested, we have provided an introduction and overview of Gja1 gene and CX43 protein function in the introduction section on pg. 4, in lines 66-70, and 78-80, and have further elaborated on the use of this gene’s promoter in reporter assays in SMCs on pg. 10, in lines 197-199. These three excerpts now read:

“For instance, the widely studied Connexin 43 (encoded by the GJA1 gene), a vital contraction-associated protein [9,10], is induced at term labor onset relative to late gestational quiescent stages, in part due to the spike in active transcription from its coding gene at term [8].”

“For example, AP-1 heterodimers, but not homodimers, tend to activate the murine Gja1 promoter [11], and this heterodimeric activation effect can be heightened or hampered depending on which major PR isoform is present in SMCs [12].”

“As previously outlined, Gja1 is necessary for the initiation of contractions [9,10] and the output of this gene promoter has often been used as a proxy for the laboring transcriptional program phenotype.” 

7) Line 173. The brief explanation of JUND and FOSL2 is also necessary here or in the Introduction section. I know that these genes are the components of Jun and Fos subfamily, respectively, but the AP-1 transcriptional factors and its related molecules should be concisely explained for readers.

As requested, we have provided this brief introduction to and explanation of JUND and FOSL2 functions on pg. 10, in lines 202-204. The text now reads:

“Within the AP-1 FOS and JUN subfamilies, FOSL2 and JUND are especially highly expressed at the protein level at labor onset in mice [19], and we therefore selected these factors for our AP-1 homodimer and heterodimer test combinations.”

8) The data in Figure 3 are very interesting. Gja1 and Fos promoter activation increased in the presence of MYB. In contrast, ELF3 suppressed the Gja1 and Fos promoter activation. This luciferase assay-construct is elegant. This is just a comment.

Thank you for this comment.

9) The paragraphs of “AP-1 factors positively auto-regulate the Fos promoter in myometrial cells” (lines 200-221) and “Fos promoter activities in the presence of MYB or ELF3 mirror the transcriptional effects observed at the Gja1 promoter” (lines 233-249) are the result of same figure, Figure 4, so it might be better to combine these two paragraphs together.

Because the effects of AP-1 factors on Fos promoter activity in laboring SMCs have not, to our knowledge, yet been tested, we wanted to highlight the importance of this finding before turning to the additional effects of MYB or ELF3 in this context, and we provided two headings for data in the same figure. Because we did not deem it appropriate to show two separate figures with partial overlap in the displayed data, we kept these data (showing the effects of AP-1 on Fos promoter activity first, then the effects of adding MYB with or without AP-1 factors) in the same figure. 

10) Overall, I recommend describing the key molecules (for example, Gja1, AP-1, Jun, Fos, etc) in the Introduction section. Most of these molecules are introduced in the Result section at present, so it is hard to follow for readers.

We thank the reviewer for their recommendations and have modified the introduction to address this comment.

RESPONSE to REVIEWER 2

Reviewer #2: SUMMARY:

The authors examined two genes, Myb and Elf3, identified to be up-regulated during mice labor in a previously published study that utilised total RNA-seq. Firstly, they correlated the myometrial tissue expression levels of their transcripts to (i) different stages of in vivo physiological pregnancy and labor in mice (d15, d17, d19, d19.5 and d20; all five time points for CD-1 and four for Bl6 strain) and humans (no labor and labor; both term gestation), and (ii) two established in vivo mouse models of preterm labor (LPS and RU486 exposure). Secondly, luciferase-based reporter gene assays were used to demonstrate the effects of Myb and Elf3 expression, alongside genes that encode FOSL2 and JUND (AP-1 monomers), from plasmids transfected into immortalised SHM cells on the activities of (iii) Gja1 and (iv) Fos gene promoters (cloned from Bl6 genomic DNA). Results show (i) previous RNA-seq findings validated using qPCR, (ii) RU486 increases Myb and Elf3 transcripts more so than LPS, and (iii)-(iv) Myb expression enhances both Gja1 and Fos gene promoter activity but the opposite was observed from Elf3 expression. Together, these preliminary findings present Myb and Elf3 as novel labor-associated transcription factors of interest. 

We would like to thank the reviewer for their assessment of our work.

SPECIFIC COMMENTS:

Abstract

-Minor:

• Lines 25-27: the first letter of each transcription factor (full) name should be lower case.

We have made the requested changes in the abstract on pg. 2, in lines 24-26.

• Line 34: use lower case for ‘lipopolysaccharide’.

We have made the requested change on pg. 2, line 34.

Introduction

Concise and well-structured; relevant references have been included.

-Major:

• Lines 83-86: it would be helpful to briefly explain why MYB and ELF3 were chosen as novel proteins of interest in the Introduction, rather than this first being provided in the Results (lines 104-105), section. Furthermore, explicit hypotheses/aims of the study should be stated to justify the choices of experiments that were undertaken.

As per our response to Reviewer #1, we have provided our justification for focusing on MYB and ELF3, and our hypotheses regarding their activities in the laboring myometrial context in the introduction section, on pgs. 5-6, lines 94-109. The text now reads:

“We hypothesized that these two proteins might act as potential regulators of the laboring SMC phenotype for several reasons: both their corresponding genes are more highly transcribed at term relative to earlier quiescent gestational stages in mouse myometrial tissues [7,8,14]. Associated predicted binding motifs for both proteins are also contained within promoters of contraction-associated genes like Gja1. More notably, MYB and ELF3 have been shown to behave as transcriptional activators in cooperation with AP-1 factors in other cellular contexts. For example, Tapias and others determined that, in HeLa cells, MYB both controls the transcription of and co-localizes with AP-1 factors at the promoter of Sp3 [15], a gene prominently expressed in SMCs [reviewed in 13]. Likewise, a study by Otero and others demonstrated that ELF3 binds the Mmp13 gene promoter in chondrocyte cells; while AP-1 heterodimers can activate the Mmp13 promoter, the addition of ELF3 further compounds this activation effect [16]. ELF3 has also been shown to interact with Nf-κβ in prostate cancer cells in response to pro-inflammatory stimuli that affect late-gestation SMCs [17]. In light of these findings, we hypothesized that MYB and ELF3 might be significant members of the laboring SMC transcription regulatory network, and ones that function in conjunction with AP-1 factors.”

-Minor:

• Line 46: please define “term” gestation for PLOS readers who may not be familiar with obstetrics terminology.

As requested, we have provided the definition of “full term” for humans. The text on pg. 3, lines 45-46, now reads: “full term (or at least 39 weeks in humans)…”

Results

-Major:

• Line 138-139: it should be explicitly stated that the modelling of ‘idiopathic’ human preterm labor using mice in the present study is based on the assumption (not definitively proven) that human idiopathic preterm labor is caused solely by loss of progesterone receptor function when using RU486.

As requested, we have provided an acknowledgment of this assumption on pg. 8, in line 166. The text now reads: “a model for ‘idiopathic’ human preterm labor, and one assumed to be caused by PR function loss [18].”

• All figures: please state in the captions what the error bars are representative of (e.g. mean ± SEM or SD).

As requested, we have updated the captions in all figures to indicate that error bars are representative of average expression values ±SD.

• Fig S2: can the authors provide data to show MYB protein abundance in transfected cells (alongside the FLAG tag abundance data shown)?

We have performed Western blots on the contents of lysates from SMCs and other MYB-expressing cells and tested several commercially available anti-Myb antibodies. For all tested antibodies, we either did not observe any band, or we observed multiple bands and a lack of a size-specific band in our transfected SHM cells as well as other control cells. As a result, we are not confident that any of these antibodies work to correctly mark the Myb protein, and are therefore not confident about submitting data from any of these antibodies for publication. The western indicating MYB-Flag abundance does show expression of the MYB-flag protein at the expected size only in transfected cells. Although this does not reveal if there is any expression of endogenous MYB in the untransfected cells, transfection of MYB-flag into SHM cells did have an effect on expression of the reporter indicating that even if there is some expression of MYB under basal conditions the transfected in MYB-flag was still able to modify gene expression.

• Fig 3 and 4: captions for these figures state “Average expression level values for each construct normalised to expression output of promoter construct subject to no TF treatment” – does this mean that cells transfected with both TF construct(s) and pG-Gja1P were all normalised to cells with only pG-Gja1P, and hence why there are no error bars for the ‘pG-Gja1P’ group? If all values from cells transfected with TF constructs were normalised to ‘no TF treatment’, does this mean that ‘no TF treatment’ values used for ANOVA were all ‘1’? Why not present (and undertake statistical analysis on) the values “calculated by normalising the firefly to Renilla expression” (line 437) without further normalisation to the ‘pG-Gja1P’ group?

To clarify, our normalization method for our luciferase assay data in this paper involves using reporter output values from the promoter construct alone to normalize the reporter outputs from every other experimental condition (i.e. promoter construct with one or more of MYB, ELF3, and AP-1 factors). In so doing, we follow a past precedent in the literature, wherein output from the promoter construct alone is commonly used to normalize each set of experimental data because there can be variation in the magnitude of the response within each individual replicate set, even though the overall profile of changes remains the same. We have therefore normalized and represented our luciferase reporter data in this way to be consistent with the ways in which these kinds of data have been presented in prior scholarship. For examples of these kinds of data representations, see Thomas et al., 2021; Thakur et al., 2019; Sinh et al., 2017; Nadeem et al., 2016; Lim and Lappas, 2014; Lim et al., 2013; Blum et al., 2012; Lindström et al., 2008; and Mitchell and Lye, 2005.

• Fig 3 and 4: please state why JUND and FOSL2 were the chosen members of the AP-1 family of proteins to examine using the reporter assay, and thus why the other AP-1 monomers appear to not be relevant to this study.

As per our response to Reviewer #1, we have provided this explanation of JUND and FOSL2 on pg. 10, in lines 202-204. The text now reads:

“Within the AP-1 FOS and JUN subfamilies, FOSL2 and JUND are especially highly expressed at the protein level at labor onset in mice [19], and we therefore selected these factors for our AP-1 homodimer and heterodimer test combinations.”

-Minor:

• Fig 1, 3, 4, S1: is it possible for the authors to add or use alternative annotations to indicate statistical significance on graphs for ANOVA? For example, in Fig 3, it is unclear whether ‘pG-Gja1P, JUND/JUND’ vs ‘pG-Gja1P, JUND/JUND/MYB’ are significantly different according to ANOVA.

The ANOVA method is the most applicable standard statistical method used to test statistically significant differences among multiple experimental conditions for the qPCR and reporter assay data in the indicated figures. This method does not, as in the case of a t-test, test the statistical significance of any one individual pair of compared conditions; rather, the ANOVA allows us to compare the means of multiple conditions to test if they are significantly different from one another before completing the multiple pairwise comparisons.

As per the letter-based representational significance display, the occurrence of the same letters between two experimental conditions represents no statistically significant differences between those conditions. In the case the reviewer has highlighted here, the addition of MYB to pG-Gja1P, JUND/JUND does not significantly alter promoter activity compared to pG-Gja1P (hence the letter “B” applies to both conditions). We have used the legends in the referred-to figures to indicate how the letter display is to be interpreted: as stated in the figure captions, “Groups determined by one-way ANOVA to be significantly different (p < 0.05) from one another are labeled with different letters; to indicate p > 0.05, groups are labeled with the same letter”

• Fig 2: check style consistency (symbol colours and error bar thickness) of ‘B’ graph on right.

We have corrected the above-mentioned consistency errors in Fig 2. All symbol colors and error bar thicknesses are uniform.

• Fig 2 caption (line 154): please give the full name for “hpi” if only used once in the manuscript; also check that the ‘B’ and ‘C’ labels for the graphs match the description in the caption.

We have given the full name for “hpi” on pg. 9, in line 183 and have rewritten the figure caption to match the (revised) figure on pg. 9, in lines 183-186. The text now reads:

“Expression levels (±S.D.) of transcription factor-encoding genes Myb (B) and Elf3 (C) in sham- and LPS-treated mice (left) and in vehicle- and RU486-treated mice (right), as determined by RT-qPCR.”

• Fig S2: please state full names for all abbreviations used to label the figure in the caption.

We have stated the full names for the abbreviations in Fig S2. The text now reads:

“Protein levels of factor encoded by designated construct, with pCDNA3.1 (pcDNA) empty vector used as negative control (CTL).”

Discussion

Good consideration of recent developments relevant to MYB and ELF3.

-Major:

• Please add a ‘Study limitations’ paragraph to briefly discuss, for example, (i) limits of using cell lines and animal models to infer expression/function-related changes during human parturition (particularly for addressing the discrepancy between in vivo gene expression and in vitro gene promoter activity data for Elf3), (ii) absence of molecular biology techniques that assess interactions between transcription factors as postulated by the authors (e.g. line 262-265), and (iii) absence of functional assays beyond transcription factor/gene promoter activity that would truly demonstrate the impact of MYB and ELF3 expression on myometrial phenotype (amongst all other changes that will occur at a multitude of other proteins during physiological and pathological labor).

As requested, we have written a separate paragraph that responds to all three listed points in our discussion section on pg. 17, in lines 359-371, where we acknowledged the three aforementioned limits and indicated the kinds of experiments that could further validate the roles MYB and ELF3 play in pregnant and/or laboring SMCs. The text now reads:

“Still further work needs to be done to uncover the exact molecular mechanisms by which MYB and ELF3 impart their activities in SMCs during labor-proximal gestational stages. Non-human cell lines and animal models offer a preliminary glance at the molecular players controlling contraction-driving genes. Establishing functional changes directly relevant to human parturition, however, requires additional experimental techniques. If MYB and/or ELF3 act as transcription factors in the same complex as AP-1 and PR factors, proof-of-interaction tests via co-immunoprecipitation or proximity ligation assay experiments in human tissues are needed. Chromatin immunoprecipitation experiments can also indicate whether either protein directly binds open chromatin regions in contraction-associated genes in human term not in labor or active laboring SMCs. More broadly, If MYB and ELF3 play an essential role in establishing the contractile physiological phenotype in the myometrium, uterine contractility tests on transcription factor overexpression models like those developed by Peavey et al. for PRs [28] can best address this question.”

• Lines 282-290: it would be good for the authors to present suggestions for experiments to be done to address the two conjectures proposed to explain high Elf3 expression in vivo despite repressive effects on gene promotor activity in vitro (to elaborate on “Further work” at line 291).

As requested, we have extended our discussion to incorporate the requested suggestions on pgs. 16-17, lines 349-351 and 355-358. The text now reads:

“The question remains as to why Elf3 is highly transcribed at labor relative to day 15, when its activity would assumedly be more needed to support a labor-associated gene repression program. We present two separate, but not mutually exclusive hypotheses: since ELF3 has been shown in different cellular contexts and processes to act as an activator or repressor, perhaps ELF3 can exert a gene-activating role during labor if the protein contains different post-translational modifications or transcription factor partners that were not present in our current study. Examining the modification status of ELF3 at non-laboring and laboring stages would give a first indication if stage-specific differences exist that might impose functional consequences. Alternatively, but not necessarily mutually exclusively, our finding might indicate a possible role for ELF3 in the early postpartum period, the onset of which requires a rapid reduction in labor-associated gene expression events. In this case, having high quantities of ELF3 readily available already during labor may assist in the successful transition from SMC contractility back to quiescence. Here, testing the effects of conditional blocking of Elf3 expression before or at the labor onset stage in SMCs in mouse models can partially address this question by offering insights as to whether the postpartum involution process is consequently hindered or significantly altered.”

• Please elaborate on the difference observed for Myb and Elf3 expression between LPS and RU486 mice preterm labor models, and explicitly state the relevance of AP-1 to the RU486 model that would further justify it being the focus of the reporter assays.

As requested, we have elaborated on our finding regarding differential expression differences for Myb and Elf3 between the two preterm labor models, and discussed the significant of AP-1 factors in the RU486 model in our discussions section on pg. 15, in lines 303-309. The text now reads:

“Our further finding that Myb and Elf3 undergo a particular spike in expression in the idiopathic, but not the infection-simulating model of preterm labor suggests that the corresponding proteins may be involved in specific pathways in laboring SMCs. MYB and ELF3 may be targeted specifically by the PR signaling pathway at term. Since AP-1 factor-mediated effects on Gja1 promoter activity differ depending upon their PR isoform partner and its ligand status [12], it is likely that other transcription factors may be likewise affected.”

Materials & Methods

-Major:

• A good description of how mice myometrium tissues were dissected and stored after removal from the body has been included (lines 362-367). Please add an equivalent description for human myometrium tissues in the ‘Human tissue collection’ section.

We have expanded our description of the human myometrial tissue collection on pg. 19, in lines 407, and 415-417. The text now reads:

“After receiving written consent, myometrial biopsies from healthy women undergoing term not in labor elective Caesarean sections (TNL, n=5) or spontaneous, non-induced term labor (TL, n=5) were collected and transferred from the operating theatre to the laboratory. For the TL group, biopsies were obtained during emergency Caesarean sections due to breech presentation and fetal distress. Patients with preterm premature rupture of membranes (PPROM), clinical chorioamnionitis, fetal anomalies, gestational diabetes/hypertension, cervical cerclage, preeclampsia, antepartum hemorrhage, and autoimmune disorders were excluded from the study. After the delivery of fetus and placenta, a small 1.5 cm sliver of myometrium was collected from the upper margin of the uterine incision made in the lower uterine segment in both laboring and non-laboring women prior to closing of the uterus. The endometrium was removed, and myometrial tissues were flash-frozen in liquid nitrogen and stored at −80°C until needed.”

• Lines 325-327: were cases of artificial induction and augmentation of labor also excluded?

Yes, we only collected myometrial samples from patients undergoing spontaneous, non-induced term labor and elective Caesarean procedures. We have clarified this in our methods paragraph on pg. 19, in line 407 (see in above response).

• Lines 375-376: either include Supplemental Data and/or citations to support the statement that “these reference genes were most consistently expressed at similar levels across gestational time points”.

As requested, we have added the citations to the studies that have previously utilized these reference genes on pg. 22, in line 466. 

• Line 377-379: review the technical accuracy and/or whether the positioning of this statement at the end of the paragraph makes sense. Were the authors referring to checking for genomic DNA contamination during qPCR, which uses DNA polymerase, or at conversion of RNA to cDNA using reverse transcriptase (and checked using agarose gel electrophoresis)?

We have provided this requested clarification of on pg. 22, in lines 467-469. The text now reads: 

“All samples were confirmed not to have DNA contamination because no target amplification was observed with DNA polymerase in the qPCR reactions for reverse transcriptase-negative samples.”

• “Reporter plasmid cloning and acquisition” section (pages 18-19): insert mention of S2 Fig, and also state whether transfection efficiency for cells used for reporter assays was checked for all n=3 shown in Fig 3 and 4.

Since S2 Fig displays the western blot results that confirm the successful transfection of SHM cells with our constructs of interest, we have added a reference to S2 Fig in the Western blot section on pg. 25 in line 522, as this appears the more appropriate location for this reference.

In first establishing our transfection experiments, we set up a test transfection with pmaxGFP to ensure that the high (>90%) transfection efficiency commonly associated with SHM cells was observed in our own experiments. When we checked for transfection with pmaxGFP, we observed both at 24 hours post transfection and at 48 hours post transfection that our results were consistent with this fact, and that the vast majority of cells (>90%) are well-transfected.

We also want to note that the Renilla (pGL4.75) plasmid we use in our luciferase vectors internally controls for transfection efficiency. The output of Renilla can vary from replicate to replicate, but is similar within each condition and therefore allows us to accurately determine, despite replicate-to-replicate variation, whether the overall profile of changes in reporter activity between different test conditions is consistent. We also noted that all experiments display a similar level of Renilla expression from replicate to replicate, indicating no gross variations in transfection efficiency.

• Table S3: check all primer sequences provided are accurate, as well as their text formatting.

As requested, we verified that all provided primer sequences are accurate, and have added a brief caption that explains the formatting of the primer sequences in S3 Table in particular. This caption reads:

“Lowercase letters represent additional portion of Gja1 coding region before the first nucleotide in the codon encoding the first Met residue (Gja1) or the overhang portion of the primer corresponding to the sequence in the target vector backbone. Refseq accession numbers obtained from portion of primer sequences that does not correspond to overhang.”

• Line 423: please state what equipment (and software) was used to acquire the (digital) images of the ECL reagent-treated membranes, as well as briefly describe how the images were analysed.

We have provided the equipment and software name details and expanded upon the membrane treatment details on pg. 25, in lines 522-523. The text now states:

“Membranes were developed and images were visualized (SFig2) using Bio-Rad Image Lab Software.”

Minor:

• Details for animal research ethics (lines 332-336) would be better placed under the same ‘Ethics statement’ heading (page 15) used for the ethics described for the human-derived samples, but as a separate paragraph. The remainder of the ‘Animal model’ section (lines 336-342) could then form the first paragraph of the ‘Mouse gestational model myometrial tissue collection’ headed section (page 16) instead.

As per the reviewer’s comments, we have reorganized these sections on pages 19-22, lines 392-455. The text with the pertinent changes now reads:

Lines 392-403: “Ethics statement 

This study was carried out in accordance with the protocol approved by the Research Ethics Board, Sinai Health System REB# 02-0061A and REB# 18-0168A. All subjects donating myometrial biopsies for research gave written informed consent in accordance with the Declaration of Helsinki. All research using human tissues was performed in a class II-certified laboratory by qualified staff trained in biological and chemical safety protocols, and in accordance with Health Canada guidelines and regulations. 

All mouse experiments were approved by the Animal Care Committee of The Centre for Phenogenomics (TCP) (Animal Use Protocol #0164H). Guidelines set by the Canadian Council for Animal Care were strictly followed for handling of mice. Virgin outbred CD-1 or inbred Bl6 (C57/Bl6) mice used in these experiments were purchased from Harlan Laboratories (http://www.harlan.com/).”

Lines 419-455: “Mouse gestational model tissue collection

All animals were housed in a pathogen-free, humidity-controlled 12h light, 12h dark cycle TCP facility with free access to food and water. Female CD-1 and Bl6 mice were naturally bred; the morning of vaginal plug detection was designated as gestational day (d) 1. Pregnant mice were maintained until the appropriate gestational time point. For term labor models, gestation in mice on average is up to 3 weeks, where delivery under these conditions occurred during the evening of d19 or morning of d20. Our criteria for labor were based on delivery of at least one pup.

In the spontaneous labor gestational model, samples were collected on gestational days 15, 17, 19 term-not-in-labor (TNIL), 19-20 during active labor (LAB), and 2–8 hours postpartum (PP) for CD-1 mice and on gestational days 15, 18.75 (TNIL), 19-20 during active labor (LAB) and 2–8 hours PP for Bl6 mice. Myometrial tissues were collected at 10 AM on all gestational days with the exception of labor samples (LAB), which were collected once the dams had delivered at least one pup. In the infection-associated inflammation preterm labor model, mice underwent a mini-laparotomy under isoflurane for anesthesia on gestational day 15, and were given an intrauterine injection of either sterile saline (sham) or 125 µg of E. coli-derived LPS (serotype 055:B5) in sterile saline. Mice were sacrificed 24 hours after sham surgery or during LPS-induced preterm labor, which occurred 24 hours post infection +/- 6 hours. In the idiopathic preterm labor model, mice were subcutaneously injected with vehicle (corn oil/ethanol) or 150 µg of mifepristone (RU486) on gestational day 15. Mice were sacrificed 24 hours after injection or during RU486-induced preterm labor, which occurred 24 hours post injection +/- 2 hours.

Mice were euthanized by carbon dioxide inhalation. The part of the uterine horn close to the cervix from which the fetus was already expelled during term or preterm labor was removed and discarded; the remainder was collected for analysis. For each day of gestation, tissue was collected from 4-8 different animals. Isolated uteri were placed into ice-cold PBS. Uterine horns were bisected longitudinally and dissected away from both pups and placentas. The decidua basalis was cut away from the myometrial tissue. The decidua parietalis was carefully removed from the myometrial tissue by mechanical scraping on ice, which removed the entire luminal and glandular epithelium and most of the uterine stroma. Myometrial tissues were flash-frozen in liquid nitrogen and stored at −80°C until needed.”

• Line 320: please add “Human” to the start of the ‘tissue collection’ heading to better distinguish from the section that describes mice tissue collection.

As requested, we have added this term on pg. 19, in line 405.

• Line 324: do the authors mean ‘breech’ where it states “bridge” presentation?

We have made this correction on pg. 19, line 409.

• Would it be possible for the authors to present median with range values for gestation age, gravida, and parity for their human study participants – potentially in a Supplemental Table? BMI, ethnicity, whether participants had a previous caesarean section, and presence/absence of fetal membrane rupture immediately prior to caesarean section would also be of interest if available.

All term not in labor and term labor samples were collected between 38-40 weeks, while a history of previous Caesarean section is part of our inclusion criteria for term not in labor samples. Aside from this, we do not have the requested information for all samples from available recordings at the time of collection.

• Line 350: insert ‘preterm’ before “labor”.

We have made this insertion on pg. 21, in line 439.

• Line 373: state what qPCR reagent was used with the cDNA samples and primers (e.g. SYBR Green?).

We have clarified that we have used SYBR Select from Thermo Fisher Scientific on pg. 22, in line 461.

• Lines 374-375: state which housekeeping gene was assigned to each mouse strain (as indicated in Fig 1 and S1).

As requested, we have made this assignation on pg. 22, in line 463. The text now reads: 

“Target gene expression was monitored by qPCR using SYBR Select (Thermo Fisher Scientific) and primers that target sequences within the exons of pertinent mRNA transcripts (S1/2 Tables) and normalized to levels of total H1f0 (Bl6 mouse), Tbp (CD-1 mouse) or Mapk1 (human) mRNA.”

• Tables S1, S2 & S3: please add RefSeq accession numbers used to confirm specificity of each pair of primers (using e.g. NCBI Primer-BLAST).

We have provided the requested accession numbers in the tables, as appropriate.

• Lines 382-383: Abbreviations – “DMEM media” technically uses the word ‘media’ twice; same applies to “Opti-MEM media” at lines 433-434. Please state the full name for “FBS” (instead of using the abbreviation if only used once in the manuscript).

We have removed the double use of the word “media” in the cell culture methodology section, and stated the full name of FBS, as requested, on pg. 22, in line 473, and pg. 25, in line 534.

• Lines 381-385: please state where the SHM cells were sourced from. Purchased commercially or donated by a named collaborator?

We have clarified our internal sourcing of SHM cells from Professor Oksana Shynlova on pg. 22, in line 472.

• Line 382: specify type of plate if “plate” is to be used in this sentence.

We have removed the term “plate” in this instance (now removed on pg. 22, in line 473).

• Line 383: state the working concentrations of penicillin and streptomycin in the media used instead of stating “1%”.

We have provided these working concentration values on pg. 22, in line 474. The text now reads: 

“Syrian hamster myometrial (SHM) cells supplied by Professor Oksana Shynlova were cultured in phenol-free DMEM supplemented with 10% fetal bovine serum, 100 IU/ml penicillin, and 100 µg/ml streptomycin (Pen-Strep).”

• Line 385: please state what method of mycoplasma testing was used.

We have clarified that we have utilized the Mycoplasma PCR Detection Kit from Biovision (catalog #K1476-100) on pg. 23, in line 477.

• Line 395: what was the “pCI” construct used for?

We have provided this clarification on pg. 23, in lines 488-492. The text now reads:

“All transcription factor-encoding constructs consisted of a pcDNA3.1 plasmid backbone, with the exception of the plasmid encoding ELF3, which consisted of a pCI plasmid backbone. The potential regulatory activity of any one transcription factor-encoding plasmid was always compared to its gene-free plasmid backbone counterpart.”

• Lines 395-397: check the relevance of references 29 and 30 for their corresponding sentences.

As requested, we have checked and corrected these references on pg. 23, in lines 494 and 496. The text now respectively refers to the original paper characterizing ELF3 and its associated vector (Oettgen et al., 1997) and the paper assigned to the MYB vector we used that is deposited with Addgene (Roe et al., 2015).

• Lines 402 and 435: missing degrees symbol in “-80C”.

We have added the degrees symbol in these instances, now on pg. 24, in line 501 and on pg. 25, in line 536.

• Line 417: please check the catalog number or supplier info for anti-ELF3.

We have checked and corrected the catalog number to its correct name, A6371, now on pg. 24, in line 516. 

Other minor points

• Check “Hist1” is the correct gene name (and not referring to a gene cluster).

As requested, regarding this gene, we have provided the name that is associated with the most updated version of the UCSC Genome Browser, both throughout the manuscript and in the relevant figure.

• ‘MAPK1’ is the official gene name for ERK2; the latter is considered a synonym 

We have changed every instance in the manuscript and relevant figure of our use of ERK2 to MAPK1.

• Check that Greek letters are used instead of their Latin substitutes (e.g. “u” (where it should be ‘�’) at lines 356 and 412).

We have made the requested corrections, now on pg. 21, in line 444, and on pg. 24, in line 511.

• Please be consistent with use of abbreviations. For example, “GD” was used to abbreviate ‘gestation day’ in line 339, but then “gestation day” was used throughout the following ‘Mouse gestational model myometrial tissue collection’ section in Materials and Methods instead of ‘GD’. Same with “pp” for ‘postpartum in lines 347-348.

We have revised the manuscript to use “d” consistently throughout the manuscript text and relevant figures to indicate the gestational day. We have made the requested alteration in the abbreviation for postpartum, now on pg. 21, in line 435.

• Please provide full name for the “TF” abbreviation at its first use (line 178?).

We have reviewed the manuscript so that the full name – transcription factor – is used consistently throughout instead of the abbreviated “TF.” As requested, we have provided the full name for “TF” in that location, now on pg. 11, in line 218.

• Check citations. For example, according to the References list, Sinh is first author for reference 20 (not 19) in lines 272-274, and cited reference 28 in line 307 looks like it should be 29.

We have checked all and corrected any erroneous citations. We would like to note that, with the addition of several new references in the revised version of the manuscript, the numbers corresponding to particular references have changed.

---

## [Decision Letter · Decision Letter 1]

6 Dec 2022

MYB and ELF3 differentially modulate labor-inducing gene expression in myometrial cells

PONE-D-22-17810R1

Dear Dr. Mitchell,

We’re pleased to inform you that your manuscript has been judged scientifically suitable for publication and will be formally accepted for publication once it meets all outstanding technical requirements.

Kind regards,

Atsushi Asakura, Ph.D

Academic Editor

PLOS ONE

Additional Editor Comments (optional):

Reviewers' comments:

Reviewer's Responses to Questions

**Comments to the Author**

1. If the authors have adequately addressed your comments raised in a previous round of review and you feel that this manuscript is now acceptable for publication, you may indicate that here to bypass the “Comments to the Author” section, enter your conflict of interest statement in the “Confidential to Editor” section, and submit your "Accept" recommendation.

Reviewer #1: All comments have been addressed

2. Is the manuscript technically sound, and do the data support the conclusions?

Reviewer #1: Yes

3. Has the statistical analysis been performed appropriately and rigorously? 

Reviewer #1: Yes

4. Have the authors made all data underlying the findings in their manuscript fully available?

Reviewer #1: Yes

5. Is the manuscript presented in an intelligible fashion and written in standard English?

Reviewer #1: Yes

6. Review Comments to the Author

Reviewer #1: The manuscript was well-revised. Additional information was included in the introduction section. I recommend this article for publishing.

7. PLOS authors have the option to publish the peer review history of their article (what does this mean?). If published, this will include your full peer review and any attached files.

Reviewer #1: No

---

## [Editor Report · Acceptance letter]

23 Dec 2022

PONE-D-22-17810R1 

MYB and ELF3 differentially modulate labor-inducing gene expression in myometrial cells 

Dear Dr. Mitchell:

I'm pleased to inform you that your manuscript has been deemed suitable for publication in PLOS ONE. Congratulations! Your manuscript is now with our production department. 

Kind regards, 

on behalf of

Dr. Atsushi Asakura 

Academic Editor

PLOS ONE